# Factors Associated with Functional Constipation among Students of a Chinese University: A Cross-Sectional Study

**DOI:** 10.3390/nu14214590

**Published:** 2022-11-01

**Authors:** Yuhan Zhang, Qian Lin, Xin An, Xiuying Tan, Lina Yang

**Affiliations:** Xiangya School of Public Health, Central South University, 110 Xiangya Road, Changsha 410078, China

**Keywords:** functional constipation, university students, dietary patterns, associated factors

## Abstract

Functional constipation (FC) is prevalent worldwide and is an increasingly prominent problem among university students. However, there is a paucity of research on FC in university students. This study aimed to assess the prevalence of FC among Chinese university students by the Rome III criteria and investigate its associated factors. This cross-sectional study was conducted by online questionnaires among 929 university students at a Chinese university. Food consumption was assessed with the Semi-Quantitative Food Frequency Questionnaire (SQFFQ) and dietary patterns were analyzed using factor analysis. A binary logistic regression model was applied to clarify FC-associated factors. The prevalence of FC among university students was 5.1%. Interestingly, among university students, the prevalence of FC with “complex” dietary pattern was significantly higher than those with “vegetable, fruit, egg and milk-based” and “livestock and aquatic product-based” dietary pattern (9.9% vs. 3.1% vs. 2.8%, *p* < 0.001). The prevalence of FC was significantly higher among university students with moderate to severe sleep disorders than those with the other sleep status (χ^2^ = 18.100, *p* < 0.001). Furthermore, after adjusting the covariates, “complex” dietary pattern (OR = 4.023, *p* < 0.001), moderate to severe sleep disorders (OR = 3.003, *p* = 0.006), overeating (OR = 2.502, *p* = 0.032), long mealtime (>30 min) (OR = 6.001, *p* = 0.007), and poor defecation habits (OR = 3.069, *p* = 0.042) were positively associated with FC among university students. Based on the above-associated factors for FC, improving dietary patterns and sleep status and developing good bowel and dietary habits are essential to prevent and alleviate university students’ FC.

## 1. Introduction

Functional constipation (FC) is a clinically common functional gastrointestinal disorder commonly appearing in children and adults worldwide [1,2]. FC, also known as chronic idiopathic constipation, refers to chronic constipation formed by dysfunction or disturbance in the physiological functions of defecation for some reasons, except irritable bowel syndrome (IBS), and without organic lesions or structural abnormality resulting in difficulty in defecation [3,4,5]. Meta-analyses showed that FC was endemic among countries and the prevalence varied across different cross-sectional surveys (i.e., different regions) [6]. According to domestic and foreign surveys, the average prevalence of constipation worldwide was 16% (between 0.7% and 79%) [7], of which the prevalence of FC diagnosed according to the Roman III criteria was 10.1% [6], and the prevalence among university students in China was 9.37% to 27.17% [8]. University students are one of the main groups of FC. If they are in a constipated situation for a long time, it will not only cause facial acne and irritability, but also lead to hemorrhoids, fissures, intestinal obstruction, and other diseases, which can affect their studies and life [9,10,11].

Existing studies have shown that a sedentary lifestyle, dietary habits and types such as low vegetable and fruit intake (low dietary fiber), inadequate water intake, and low levels of education all contribute to an increased prevalence of FC [7,12,13]. In addition to the known factors described above, the prevalence of FC may be associated with sleep status. A survey by the World Health Organization revealed that 27% of the world’s people had sleep problems [14]. Based on the “2020 China University Students Health Survey Report”, 77% of university students had experienced sleep disturbances in the past year. Sleep problems directly affect people’s lives. Current studies have found that people with gastrointestinal diseases or symptoms had high levels of sleep problems [15]. It has also been found that both sleep and the circadian cycle affected gastrointestinal function [16]. Gastrointestinal function or disease is closely related to sleep, and the relationship between FC and sleep will be explored in our study.

Rajindrajith et al. found that children or adolescents with FC had more emotional and behavioral problems [17,18]. University students are in a special stage of transition between campus life and social life, as well as an important stage of physical and mental development. They will inevitably encounter some events in their lives that are affected by their emotions in different ways, which are called life events, including positive life events and negative life events (also known as stressful life events) [19]. Some stressful life events, such as unsatisfactory examination results, disputes with close friends, loss of love, and prolonged absence from family, are more likely to lead to the emergence of negative emotions and behaviors in university students [20], which may lead to gastrointestinal dysfunction, making FC more likely to occur.

In recent years, there has been an increasing amount of FC studies on different populations at home and abroad, but there are fewer studies on FC and its associated factors in university students. The treatment of FC is extremely challenging [21]. Therefore, the study aimed to investigate the prevalence of FC among Chinese university students using the Rome III criteria and explored its associated factors, to prevent FC or improve its status.

## 2. Materials and Methods

### 2.1. Ethical Approval

This study has been approved by the Ethics Committee of the Xiangya School of Public Health of Central South University (XYGW-2019-032).

### 2.2. Study Design and Participant Eligibility

From April to July 2019, at Central South University in Changsha, Hunan, south-central China, a cross-sectional study was conducted among university students by online questionnaires.

Participant inclusion criteria were as follows: freshman to fifth year undergraduates aged ≥18 who voluntarily participated and signed informed consent forms. The exclusion criteria were as follows: (1) those who have digestive system diseases, hematologic diseases, and chronic cardiovascular system diseases; (2) those with serious lesions in other organs.

### 2.3. Sample Size Estimation

According to research studies, the prevalence of FC among university students in China ranged from 9.37% to 27.17% [8]. From the cross-sectional study (tolerance error of 0.2 p), the required sample size is between 257 and 929. Considering the 10% non-response rate and human, material, and financial factors, a total of 1000 questionnaires were distributed in this survey. A total of 950 university students participated in the survey and 935 completed the questionnaire, of which 929 university students (effective recovery rate of 97.8%) were the final valid samples (see Figure 1).

### 2.4. Data Collection

The online questionnaires were completed by university students using Questionnaire Star through various electronic devices under the guidance of uniformly trained investigators. Questionnaire Star is an online questionnaire tool for creating, submitting, and collecting questionnaire information.

Demographic Information: University students’ gender, age, grade, height, and weight.Diagnostic Criteria for FC: According to “the Diagnosis and Treatment of Functional Constipation” [22], the Rome III criteria were used to determine whether university students had FC or not, which is used for presenting the symptoms at least 6 months before diagnosis and meets the criteria in the past 3 months.Information on Lifestyle Habits: University students’ poor defecation habits (playing with mobile phones and reading books during defecation), common means of transportation, dietary habits (overeating, mealtime), and drinking water (daily water intake, active drinking water).Physical Activity Evaluation Criteria: The International Physical Activity Questionnaire (IPAQ) was used to investigate the physical activity levels of university students over the past week, which has already shown good reliability and validity in Chinese college students [23]. The IPAQ short-scale consists of 7 question entries. According to the IPAQ short-scale scoring criteria, the physical activity level of university students was divided into three levels, i.e., light, moderate, and vigorous physical activity.Sleep Status Evaluation Criteria: The Self-Rating Scale of Sleep (SRSS) [24] was used to evaluate the sleep status of university students in the past month. The scale has good reliability (Cronbach’s alpha coefficient = 0.6418) and validity (r = 0.5625) [24]. According to the 10 items of SRSS, the total score ranges from 10 to 50 points, and the higher the total score, the more the sleep problem and the worse the sleep status. In this study, 10–19 is classified as good sleep status; 20–21 as fair sleep status; 22–25 as mild sleep disorders; and 26–50 as moderate to severe sleep disorders.Evaluation Criteria for Stressful Life Events: The Adolescent Self-Rating Life Events Check List (ASLEC) [25] was used to assess the frequency and intensity of stressful life events in university students in the past 12 months, which is composed of 27 items and can be classified into 6 factors: interpersonal relationships, learning stress, punishment, loss, health adaptation, and others. The scale has good reliability (Cronbach’s alpha coefficient = 0.92) [25]. A 6-level score was used, from “not occurring = 0 points” to “extremely heavy impact = 5 points”. The higher the score, the greater the impact of stressful life events, and the higher the degree of psychological stress.Dietary Pattern Analysis: The Semi-Quantitative Food Frequency Questionnaire (SQFFQ) [26] was used to collect information on the frequency and consumption of various food by university students over the past six months. Through the pre-survey, we aimed to understand the most common food varieties eaten by university students; some of the same types of food varieties were clustered, and the 22 kinds of foods obtained formed the food list of the SQFFQ. Factor analysis was used to evaluate and classify the dietary patterns of university students. According to Kaiser standards, the extracted principal factors were those with eigenvalues greater than one. Varimax orthogonal rotation was used to ensure that the factor structure was practically meaningful.a. The diet frequency of each food or food group was recorded into the number of times per week, e.g., 3 times a day or more = 21 times/week, 2 times a day = 14 times/week, and 1 time a day = 7 times/week.b. Intake of each type of food per time: (i) solid foods: 250 g, 200 g, 150 g, 100 g, and 50 g; (ii) liquid foods: 250 mL, 200 mL, 150 mL, 100 mL, and 50 mL.The Dietary Diversity Score (DDS) was assessed by trained researchers with a nutritional background using the Food Frequency Questionnaire (FFQ) [27]. Based on the dietary structure and habits of university students, their daily diet was classified into nine food groups, including cereals and potatoes, starchy staples, vegetables, fruits, livestock meat, aquatic products, eggs and milk, beans and nuts, and mushrooms. For any food group consumed once a week, a score of 1 was registered, with a DDS of 0–9. The higher the DDS, the more diverse the diet. In this study, a threshold less than 5 was defined as low DDS.

### 2.5. Statistical Analyses

The questionnaires were entered using Epidata 3.0 software (The Epi Data Association, Odense, Denmark), double-entry; statistical analysis using IBM SPSS 26.0 software (IBM Corp., Armonk, NY, USA) was performed. The test level α = 0.05 and *p* < 0.05 is statistically significant. The data meet the normality and variance homogeneity, with means and standard deviations or numbers and percentages used for the basic characteristics; Student’s *t*-tests and chi-square tests were used to analyze the relationship between FC and single-associated factors. Factor analysis and binary logistic regression models were used to determine dietary patterns and analyze associated factors, respectively.

## 3. Results

### 3.1. Participant Characteristics

A total of 929 university students were enrolled in this study. Of them, 355 (38.2%) were male and 574 (61.8%) were female, mainly freshmen and sophomores, aged 18 to 21 years old, with mostly normal BMI (65.0%). There were no significant differences among university students of different genders across grades and age groups (*p* < 0.05). The prevalence of FC in university students was 5.1%, with 70.2% of females and 29.8% of males (See Table 1).

### 3.2. Study on the Association between FC and Lifestyle Habits, Physical Activity, Sleep Status, and Stressful Life Events among University Students

As shown in Table 2, significant differences in the prevalence of FC were found among different dietary habits of university students. The higher the frequency of overeating, the higher the prevalence of FC (3.4% vs. 5.9% vs. 9.6%, *p* = 0.018). The prevalence of FC was higher among university students with long meal times (2.6% vs. 6.1% vs. 13.3%, *p* = 0.007). There were no significant differences in the prevalence of FC among university students in terms of poor defecation habits (playing on mobile phones and reading books during defecation), common means of transportation, drinking water, and physical activity (*p* > 0.05).

The SRSS scale showed that the prevalence of FC was significantly higher among university students with moderate to severe sleep disorders than those with other sleep statuses (χ^2^ = 18.100, *p* < 0.001). The ASLEC scale showed that there are significant differences in health adaptation factor score (*p* < 0.05) and other scores (*p* < 0.05) between university students with and without FC (See Table 2).

### 3.3. Association between Diet and FC among University Students

#### 3.3.1. Dietary Pattern Analysis of University Students

The adaptability test results of the factor analysis: KMO = 0.950; Bartlett’s test of sphericity χ^2^ = 10,200.415, *p* < 0.001. The results of factor analysis and the obtained scree plots showed that the eigenvalues of the first three principal components in this study were 9.273, 1.544, and 1.352, respectively, which explained 42.152%, 7.017%, and 6.144% of the total data variance after varimax orthogonal rotation. Therefore, the cumulative contribution of the first three principal components of this study was 55.313%.

Foods or food groups with absolute values of factor loadings ≥0.48 on a certain principal component were considered to be well represented on that principal component. The first dietary factor contains aquatic products, wine, porridge, flours, potatoes, coarse grains, stuffing, processed meat products, soy products, mushrooms, fried foods, sweets, nuts, and sweetened beverages with high loadings, which is named the “complex” dietary pattern due to a wide variety of foods; the second dietary factor has high loadings of dark and light vegetables, fruits, dairy products, and eggs, with high dietary fiber and high protein, lacking staple foods and meat, which belongs to the “vegetable, fruit, egg and milk-based” dietary pattern; in the third dietary factor, the loadings of aquatic products, red meat, poultry, and rice were high, with aquatic products and meat as the main food items, so it is named “livestock and aquatic product-based” dietary pattern. Therefore, the three types of dietary patterns are named “complex”, “vegetable, fruit, egg and milk-based” and “livestock and aquatic product-based” (see Table 3).

#### 3.3.2. Association between Dietary Patterns and FC among University Students

Among the 929 university students, 284 (30.6%) were “complex”, 292 (31.4%) were “vegetable, fruit, egg and milk-based” and 353 (38.0%) were “livestock and aquatic product-based”. FC was suffered by 28 out of 284 university students on the “complex” dietary pattern (9.9%) and 9 out of 292 university students on the “vegetable, fruit, egg and milk-based” dietary pattern (3.1%). Among the 353 university students on the “livestock and aquatic product-based” dietary pattern, 10 had FC (2.8%). The prevalence of FC in the “complex” dietary pattern was significantly higher than that of the “vegetable, fruit, egg and milk-based” dietary pattern and the “livestock and aquatic product-based” dietary pattern (*p* < 0.001) (see Table 4).

Of the 929 university students enrolled in the survey, 19 (2.0%) were judged to have low DDS and 910 (98.0%) had high DDS. High DDS accounts for the majority of university students, and the prevalence of FC with low DDS was slightly higher than that of high DDS, but not statistically significant (*p* > 0.05) (see Table 4).

### 3.4. Multifactorial Analysis of FC-Associated Factors among University Students

A binary logistic regression model was used to explore the association between FC and its associated factors among university students (see Table 5). After adjusting for age, sex, and BMI, among university students, poor defecation habits (playing with mobile phones and reading books during defecation) (OR = 3.069, 95% CI: 1.042~9.043, *p* = 0.042) and overeating (OR = 2.502, 95% CI: 1.085–5.773, *p* = 0.032) were positively associated with FC; moderate to severe sleep disorders had a positive association with FC compared with good sleep status (OR = 3.003, 95% CI: 1.379–6.539, *p* = 0.006); compared with the “livestock and aquatic product-based” dietary pattern, the “complex” dietary pattern was positively correlated with FC (OR = 4.023, 95% CI: 1.871–8.647, *p* < 0.001); there was a positive association between long meal time (>15 min) and FC compared with meals within 15 min (OR = 2.769, 95% CI: 1.279–5.993, *p* = 0.010; OR = 6.001, 95% CI: 1.620–22.224, *p* = 0.007).

## 4. Discussion

In this study, we found that the prevalence of FC among university students in Changsha, China, was 5.1%. It is noteworthy that the prevalence of FC among university students was associated with dietary patterns, eating behaviors and habits, defecation habits, sleep status, and stressful life events. This study will provide clues and a theoretical basis for the prevention and improvement of FC among university students.

Numerous epidemiological investigations have shown an association between diet and constipation, focusing mainly on the effects of individual nutrients or foods and food groups [28,29]. According to our survey, 98% of university students accounted for high DDS, indicating that the majority of university students consumed a wide variety of foods and had a diverse diet. At the same time, we found that the dietary patterns of university students can be classified into three categories using factor analysis: “complex” dietary pattern, “vegetable, fruit, egg and milk-based” dietary pattern, and “livestock and aquatic product-based” dietary pattern. Current evidence from a large number of research suggests that dietary diversity is not necessarily beneficial for health or optimal dietary patterns, and can also be associated with higher energy intake and suboptimal diet patterns [30,31,32]. Our study showed that the prevalence of FC among university students with the “complex” dietary pattern was significantly higher than with the “vegetable, fruit, egg and milk-based” and “livestock and aquatic product-based” dietary patterns. This may be due to the variety of foods in the “complex” dietary pattern, which includes not only aquatic products, coarse grains, mushrooms, soy products, porridge and nuts, but also high-fat, high-energy snacks such as fried foods, processed products, wine, and sugary products such as desserts and sweetened beverages with high correlations. Several studies have shown that individuals who prefer high-fat foods, junk snacks, fried foods, or coffee, alcohol, and spices had a higher prevalence of gastrointestinal symptoms [33,34]. Rollet et al. showed a positive correlation between the occurrence of constipation and sugary products and higher energy intake [29]. When high-fat, high-energy foods, as well as sugary foods and alcohol, are consumed at high levels, a high prevalence of FC is more likely to occur, which is in line with our study results.

Furthermore, numerous studies have shown that increasing dietary fiber intake can significantly increase the frequency of stools and relieve constipation in patients with constipation, which is beneficial for gastrointestinal health [29,35,36,37,38]. Other observational studies have also reported that dairy products such as cheese and milk, as well as foods such as meat and eggs, had beneficial effects on constipation [39,40]. Rollet et al. also found that the occurrence of constipation was inversely associated with lipid and total fat intake [29]. Similar results were found in our study among university students, namely, that university students with the “vegetable, fruit, egg and milk-based” dietary pattern of high-fiber foods such as vegetables and fruits, and quality protein foods such as eggs and milk, and the “livestock and aquatic product-based” dietary pattern based on fish and meat rich in unsaturated fatty acids had a lower prevalence of FC. The above reflects the complexity of the effect of dietary patterns on FC, and the effect of dietary patterns on FC in university students can be further evaluated in the future.

In addition, we found that university students who overeat had a higher prevalence of FC. Overeating is a poor eating habit that refers to abnormal behavior of swallowing large amounts of food in a short period without restraint, violently and urgently [41]. Modern medicine has confirmed that overeating, in addition to causing weight gain, can directly exert great pressure on the gastrointestinal digestive system, resulting in gastrointestinal dysfunction, and leading to a series of gastrointestinal diseases [42], which is in agreement with our findings. The higher prevalence of FC among university students with longer meal times may be due to inattentiveness during meals, such as playing on mobile phones, or playing and chatting with people around them during meals, which can lead to digestive disorders in the gastrointestinal tract. It is also possible that university students with FC have poor appetite and prolong meal time. Based on the above results, it is recommended that university students, especially those with FC, modify dietary patterns or structures through nutritional interventions and concentrate on chewing and swallowing slowly during meals to avoid overeating to prevent and improve gastrointestinal problems.

Internet terms such as “night owl” and “senior stay-up party” have gradually become synonymous with some university students, and the sleep of university students has attracted much attention. Sleep, as a necessary process for living organisms, plays an important role in maintaining the body’s physiological functions and is an indispensable part of health [14,43]. In 350 BC, Astoria elaborated in his book *Sleep and Insomnia* that sleep is caused by hot steam produced by the stomach during digestion [44]. Some studies have shown a two-way link between chronic constipation and sleep quality, i.e., sleep disturbances may affect bowel function and increase the risk of gastrointestinal disorders, and constipation may also affect sleep quality [45,46]. In our study, the same results were found in university students, i.e., there was an association between FC and sleep status in university students, with those suffering from FC experiencing poor sleep status. This may be due to the use of electronic devices such as mobile phones for various online entertainment activities before bedtime, such as chatting online, playing games, and shopping online, which increases screen time (ST) and takes up sleep time [47]. Moreover, various pressures originating from academic, employment, and interpersonal interactions may also result in shortened sleep duration, decreased sleep quality, and sleep disorders [48], which may lead to intestinal dysfunction and be prone to FC. It may also lead to shorter sleep duration and reduced sleep quality due to suffering from FC, leading to a vicious cycle. University students are encouraged to reduce the use of bedtime electronic devices by setting their mobile phones to sleep modes, switching them off, etc. during the period from going to bed to closing their eyes. They can also alleviate bedtime anxiety by soaking feet, listening to light music, and reading books, thereby improving sleep status and preventing FC.

In this study, we also found that FC among university students was associated with stressful life events that had occurred within the past 12 months. Their life stresses mainly stemmed from health adaptation events such as significant changes in their life schedules, poor physical condition, some frustrating events, and other events such as boredom with school, lost love, and arguments or fights with others. These negative life events tend to lead to higher levels of psychological stress and emotional abnormalities among university students, which can adversely affect mental health [49]. Studies in the United States and Korea have shown that negative emotions like anxiety and depression were associated with the occurrence of gastrointestinal disorders, and an association between emotions and specific defecation habits has been suggested [50,51], which is consistent with the findings of our study. In addition, university students with FC may also experience negative emotions such as anxiety due to FC. Therefore, paying attention to the life stress and spiritual and psychological health of university students is one of the important measures to prevent and treat FC.

This study highlights the important associations between FC and diet, lifestyle behavioral habits, sleep status, and negative life events among university students. The study was conducted at Central South University, located in south-central China. With more than 34,000 university students, it is one of the universities with the largest number of university students in China. It is a typical Chinese comprehensive university with certain representativeness. However, a cross-sectional study was used in this study, and further longitudinal studies will be needed in the future to find causal relationships between FC and these factors. In addition, this study was conducted in Changsha, Hunan, and found the prevalence of FC among university students to be 5.1%, while other studies have found the prevalence of FC among university students in Fujian to be 27.17% [52], 5.45% in Shandong [53], and 11.6% in Tunisia [54]. The differences in prevalence across countries and regions may be due to the influence of factors such as regional diet and lifestyle changes, which have certain limitations in extrapolation. The differences need to be elucidated through further research.

University students are beginning to manage their own lives and health independently and they are different from the general population in terms of age, diet conditions (such as school canteens, takeaways), living environment, work and rest time, as well as the pressure of study, life, and employment, which makes university students a unique group. The present study investigated the prevalence of FC for university students and further explored FC-related factors, improving its current situation from the aspects of nutritional intervention, lifestyle, and psychological status, to enhancing the learning and life quality of university students.

## 5. Conclusions

In this study, we found that the prevalence of FC among university students in Hunan, China, was 5.1%. “Complex” dietary pattern, moderate to severe sleep disorders, overeating, long meal times, and poor defecation habits (playing with mobile phones and reading during defecation) were positively associated with the prevalence of FC among university students. Attention to the dietary patterns and habits, sleep status, life stress, and mental health of university students is crucial to preventing and improving their FC. Furthermore, prospective studies are needed to verify the causal relationship between FC among university students and its associated factors.

## Figures and Tables

**Figure 1 nutrients-14-04590-f001:**
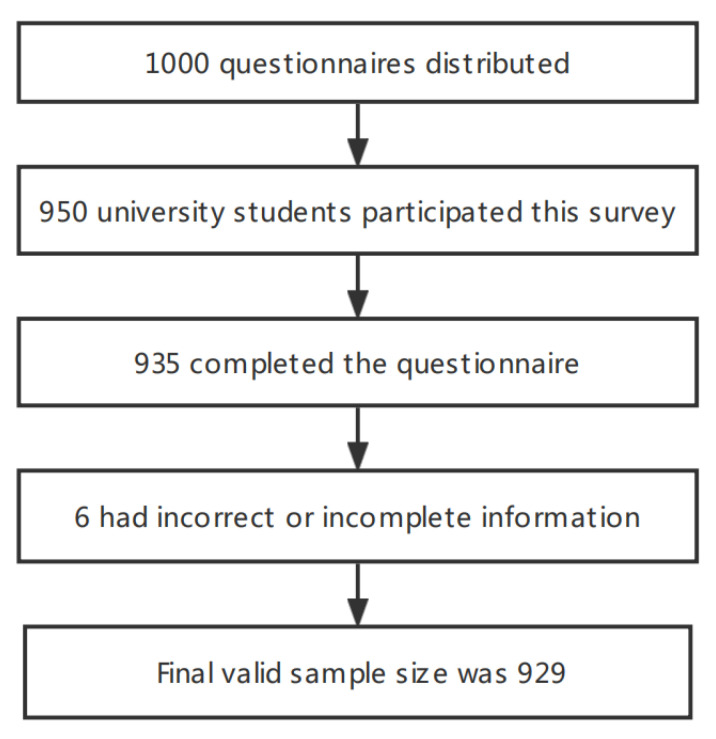
Participants enrolled according to inclusion and exclusion criteria and questionnaire completion.

**Table 1 nutrients-14-04590-t001:** Different demographic characteristics of university students.

Variables	Total (n, %)	Male (n, %)	Female (n, %)
Total	929	355 (38.2)	574 (61.8)
Grade			
Freshman	337 (36.3)	134 (39.8)	203 (60.2)
Sophomore	366 (39.4)	126 (34.4)	240 (65.6)
Junior	127 (13.7)	47 (37.0)	80 (63.0)
Senior and above	99 (10.7)	48 (48.5)	51 (51.5)
Age			
18~19	421 (45.3)	157 (37.3)	264 (62.7)
20~21	408 (43.9)	160 (39.2)	248 (60.8)
22~23	83 (8.9)	28 (33.7)	55 (66.3)
24~25	17 (1.8)	10 (58.8)	7 (41.2)
BMI			
Underweight	191 (20.6)	57 (29.8)	134 (70.2)
Normal	604 (65.0)	219 (36.3)	385 (63.7)
Overweight	69 (7.4)	31 (44.9)	38 (55.1)
Obesity	65 (7.0)	48 (73.8)	17 (26.2)
Functional Constipation FC			
FC	47 (5.1)	14 (29.8)	33 (70.2)
No FC	882 (94.9)	341 (38.7)	541 (61.3)

Age: (years old). BMI: Classified by Asian standards. FC: Diagnosed by Roman III criteria.

**Table 2 nutrients-14-04590-t002:** Comparison of behavioral habits, physical activity, sleep status, and life events between university students with and without FC.

Variables	Total (n, %)	FC $\left(n,\;\%/\bar{x}\;\pm \;s\right)$	No FC $\left(n,\;\%/\bar{x}\;\pm \;s\right)$	t/χ^2^	*p*
Poor defecation habits				3.340	0.068
No	173 (18.6)	4 (2.3)	169 (97.7)		
Yes	756 (81.4)	43 (5.7)	713 (94.3)		
Common means of transportation				0.930	0.628
Self-driving	8 (0.9)	1 (12.5)	7 (87.5)		
Bus or taxi or electric car	280 (30.1)	14 (5.0)	266 (95.0)		
Bike or walk	641 (69.0)	32 (5.0)	609 (95.0)		
Dietary habits					
Overeating				8.090	0.018 *
Hardly	473 (50.9)	16 (3.4)	457 (96.6)		
Sometimes	341 (36.7)	20 (5.9)	321 (94.1)		
Often	115 (12.4)	11 (9.6)	104 (90.4)		
Mealtime				9.802	0.007 **
0~15 min	343 (36.9)	9 (2.6)	334 (97.4)		
15~30 min	556 (59.8)	34 (6.1)	522 (93.9)		
>30 min	30 (3.2)	4 (13.3)	26 (86.7)		
Drinking water					
Active drinking water				4.294	0.117
Hardly	95 (10.2)	9 (9.5)	86 (90.5)		
Sometimes	371 (39.9)	17 (4.6)	354 (95.4)		
Often	463 (49.8)	21 (4.5)	442 (95.5)		
Daily water intake				0.484	0.785
0~500 mL	182 (19.6)	10 (5.5)	172 (94.5)		
500~1000 mL	386 (41.6)	21 (5.4)	365 (94.6)		
1000 mL and above	361 (38.9)	16 (4.4)	345 (95.6)		
Physical activity				0.190	0.909
Light	382 (41.1)	19 (5.0)	363 (95.0)		
Moderate	431 (46.4)	23 (5.3)	409 (94.7)		
Vigorous	116 (12.5)	5 (4.3)	110 (95.7)		
SRSS				18.100	<0.001 ***
Good	310 (33.4)	11 (3.5)	299 (96.5)		
Fair	160 (17.2)	5 (3.1)	155 (96.9)		
Mild sleep disorders	256 (27.6)	9 (3.5)	247 (96.5)		
Moderate to severe sleep disorders	203 (21.9)	22 (10.8)	181 (89.2)		
ASLEC					
ASLEC Score		39.98 ± 31.03	32.61 ± 26.47	−1.843	0.066
Interpersonal relationship		8.49 ± 5.72	7.55 ± 5.48	−1.090	0.276
Learn stress		8.49 ± 5.72	7.72 ± 4.98	−1.025	0.306
Punishment		7.40 ± 9.38	5.48 ± 7.77	−1.640	0.101
Loss		4.09 ± 4.58	3.06 ± 3.99	−1.693	0.091
Healthy adaptation		6.36 ± 4.24	5.01 ± 3.70	−2.421	0.016 *
Others		5.19 ± 4.59	3.79 ± 4.03	−2.310	0.021 *

* *p* < 0.05, ** *p* < 0.01, *** *p* < 0.001. Physical activity: classified by the IPAQ short-scale scoring criteria. SRSS score: 10–19 for good sleep status; 20–21 for fair sleep status; 22–25 for mild sleep disorders; and 26–50 for moderate to severe sleep disorders.

**Table 3 nutrients-14-04590-t003:** Distribution of factor loadings for each food or food group across the three dietary patterns of university students.

Foods or Food Groups	Correlation Coefficients (n = 929)		
	Complex	Vegetable, Fruit, Egg and Milk-Based	Livestock and Aquatic Product-Based
Aquatic products	0.625		0.502
Wine	0.735		
Rice			0.654
Porridge	0.603		
Flours	0.556		
Potatoes	0.671		
Coarse Grains	0.772		
Stuffing	0.529		
Eggs		0.522	
Red meats			0.658
Poultry			0.601
Processed meat	0.632		
Soy products	0.490		
Dark vegetables		0.744	
Light-colored vegetables		0.639	
Mushrooms	0.514		
Fried food	0.632		
Desserts	0.596		
Nuts	0.677		
Fruits		0.710	
Dairy		0.623	
Sweetened beverages	0.517		

The absolute value of the factor loadings <0.48 is excluded.

**Table 4 nutrients-14-04590-t004:** Association between dietary patterns, DDS, and FC among university students.

Variables	Total (n, %)	FC (n, %)	No FC (n, %)	χ^2^	*p*
Dietary Pattern				19.641	<0.001 ***
Complex	284 (30.6)	28 (9.9)	256 (90.1)		
Vegetable, fruit, egg and milk-based	292 (31.4)	9 (3.1)	283 (96.9)		
Livestock and aquatic product-based	353 (38.0)	10 (2.8)	343 (97.2)		
DDS				0.002	0.967
Low DDS	19 (2.0)	1 (5.3)	18 (94.7)		
High DDS	910 (98.0)	46 (5.1)	864 (94.9)		

*** *p* < 0.001. DDS: low DDS score < 5; high DDS score ≥ 5. Dietary patterns: obtained by factor analysis, including “complex”, “vegetable, fruit, egg and milk-based”, and “livestock and aquatic product-based”.

**Table 5 nutrients-14-04590-t005:** A binary logistic regression analysis of factors associated with FC among university students.

Variables	B	Wald χ^2^	OR (95% CI)	*p*
Poor defecation habits	1.121	4.139	3.069 (1.042, 9.043)	0.042 *
Sleep status (Good = ref)				
Fair	−0.262	0.217	0.770 (0.256, 2.315)	0.641
Mild sleep disorders	−0.117	0.061	0.890 (0.353, 2.243)	0.804
Moderate to severe sleep disorders	1.100	7.675	3.003 (1.379, 6.539)	0.006 **
Meal patterns (livestock and aquatic product-based = ref)				
Complex	1.392	12.709	4.023 (1.871, 8.647)	<0.001 ***
Vegetable, fruit, egg and milk-based	0.156	0.107	1.169 (0.460, 2.973)	0.743
Mealtime (0~15 min = ref)				
15~30 min	1.018	6.680	2.769 (1.279, 5.993)	0.010 **
>30 min	1.792	7.196	6.001 (1.620, 22.224)	0.007 **
Overeating (Hardly = ref)				
Sometimes	0.503	1.973	1.654 (0.820, 3.338)	0.160
Often	0.917	4.625	2.502 (1.085, 5.773)	0.032 *

* *p* < 0.05, ** *p* < 0.01, *** *p* < 0.001. Adjusting by age, sex, and BMI. SRSS score: 10–19 for good sleep status; 20–21 for fair sleep status; 22–25 for mild sleep disorders; and 26–50 for moderate to severe sleep disorders. Dietary patterns: obtained by factor analysis, including “complex”, “vegetable, fruit, egg and milk-based” and “livestock and aquatic product-based”.

## Data Availability

The data supporting the findings of this study are not publicly available due to the data containing information that could compromise the privacy of the participants. However, it is available from the corresponding authors upon reasonable request.

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
