# Peer review of "Factors Associated with Functional Constipation among Students of a Chinese University: A Cross-Sectional Study"

_nutrients, 2022, doi:10.3390/nu14214590_

Round 1
Reviewer 1 Report
1. Title: The title can be left as a question mark or changed as follow: “Factors associated with functional constipation among college students: a cross-sectional study” (optional change). A change in the title (not optional) would be necessary to help identify that it is a study conducted in a Chinese university: “Factors associated with functional constipation among students of a Chinese University: a cross-sectional study”.
2. Abstract: it seems appropriate to the research carried out, but again it must be clearly indicated where it was carried out (in a Chinese University). The directionality of the results and conclusion "were risk factors for university students’ FC" should be modified, since there may be reverse causality (see comments in results, discussion and conclusions sections).
3. Introduction: it is adequate, it provides the necessary information to understand the problem, its magnitude, its impact and the need to study the associated factors.
4. Objective: the authors indicate that this is the objective: “the study aimed to investigate the prevalence of FC among Chinese university students using the Rome III criteria and to explore its associated factors, to prevent FC or improve its status” . Although they have taken a large and adequate sample according to the sample calculation, it is most likely that the sample is not representative (possibly it suffers from a volunteer bias, and since it is carried out in a single University, the conclusions cannot be generalized). . to the rest, an aspect that should be noted in the limitations of the study and taken into account in its interpretation (in the discussion section).
5. Methods - study design and participants: well described.
6. Methods – settings: it is necessary to describe in more detail the location where the research is carried out, as well as the relevant dates on which the research is carried out and its different stages.
7. Methods – data collection: generally well described, but missing: a citation for the “Star Questionnaire” tool, if the “Information on lifestyle habits” was obtained through a validated questionnaire, a citation for the “International Questionnaire of physical activity” (IPAQ) short scale and a description of the domains it has and the number of items, the same for “The Self-Rating Scale of Sleep (SRSS), including the study that allows to establish validity and reliability”, which same for “Adolescent Self-Qualification of life events Check List (ASLEC)”, “Semi-quantitative Food Frequency Questionnaire (SQFFQ)” and the “Dietary Diversity Score”.
8. Results – Participants: seems well described.
9. Results - Association between FC and other variables: I do not fully understand the numbers comparison (3.5% vs 3.1% vs 3.5% vs 10.8%, p<0.001) offered in the following sentence “ The SRSS scale showed that the prevalence of HF was significantly lower among university students with good sleep status than those with sleep disorders (3.5% vs 3.1% vs 3.5% vs 10.8%, p< 0.001).” The numbers comparison should give 'college students with good sleep status' versus 'those with sleep disturbance', but four comparisons are given.
10. Results - Multifactorial Analysis of Factors Associated with FC in University Students: it is stated that bad defecation habits were risk factors for FC; that moderate to severe sleep disturbances may increase the risk of FC compared with good sleep status; Compared with the "aquatic and livestock based" dietary pattern, the "complex" dietary pattern may increase the prevalence of FC, higher the risk of FC after eating for longer periods, and overeating was a contributing factor. risk for FC. However, the study design does not allow to establish this directionality in the interpretation (due the possibility of reverse causality). For example, it is possible that people with FC spend more time on their mobile phones because they have to spend more time in the bathroom, it is possible that it is the FC that interferes with sleep, it is also possible that people with FC have less appetite and lengthen the time between meals, etc. Modify the way you report the results, indicating whether or not there is an association, and in the discussion, interpret that association considering all scenarios (including reverse causality).
11. Discussion: (a) it should be indicated if the characteristics of the sample are similar (at least) in gender and distribution in the four years of undergraduate studies (this will help to understand the degree of representativeness of the sample, since non-representativeness is assumed a priori) ; (b) it should be noted that FC prevalence should be interpreted with caution, given the impossibility of generalization; (c) interpret the results of your research taking into account all scenarios, including the possibility of reverse causality (especially in the cases suggested in the comments above).
12. Discussion – limitations: the authors should highlight the strengths of the study and its weaknesses and limitations (see previous comments: representativeness, possibility of generalization, reverse causality in some associations, etc).
13. Conclusions: this sentence should be modified: "Complex" dietary pattern correlated with sugary products and high-energy foods, moderate to severe sleep disorders, overeating, long meal times, and poor defecation habits (playing with mobile phones and reading during defecation) were risk factors for FC among university students". Some of these associations can be due to reverse causality, you may be able to draw conclusions about the direction of the associations that seem most likely to you, but for other associations the directionality may be reversed (very likely). The conclusions cannot remain like this.
Author Response
Dear Reviewer,
We sincerely appreciate your insightful and valuable comments and suggestions. You help us a lot in revising and improving our paper. Here we submit a new version of our manuscript, titled “Factors Associated with Functional Constipation among Students of A Chinese University: A Cross-Sectional Study” (Manuscript ID: Nutrients-1998348), which has been revised according to the suggestions. Please see the attachment for details of the revisions.
If you have any questions about this manuscript, please do not hesitate to contact me. We would thank you again for giving me so many precious suggestions to make our study more comprehensive and more rigorous, and make our results more convincing.
Corresponding author: Lina Yang Ph.D.
E-mail: ylnly1997@csu.edu.cn
Sincerely,
Lina Yang

Reviewer 2 Report
Dear authors,
Thank you very much for your well done article with the title „What Factors are Associated with Functional constipation among University Students?“
I have two comments:
1. At the tables 1, 2 and 4 you give your numbers in percentage as well. Please indicate, from what or in which relation the percentages are figured.
2. In the abstract, the methods, the discussion and conclusion you talk about „stressful life events“. In the results I cannot find any data to this topic, neither in the text nor the table. Please add!
Author Response
Dear Reviewer,
We sincerely appreciate your insightful and valuable comments and suggestions. You help us a lot in revising and improving our paper. Here we submit a new version of our manuscript, revised according to the suggestions, titled “Factors Associated with Functional Constipation among Students of A Chinese University: A Cross-Sectional Study” (Manuscript ID: Nutrients-1998348). Please see the attachment for details of the revisions.
If you have any questions about this paper, please do not hesitate to contact me.
Corresponding author: Lina Yang Ph.D.
E-mail: ylnly1997@csu.edu.cn
Sincerely,
Lina Yang
